# Comparative Study on Curcumin Loaded in Golden Pompano (*Trachinotus blochii*) Head Phospholipid and Soybean Lecithin Liposomes: Preparation, Characteristics and Anti-Inflammatory Properties

**DOI:** 10.3390/molecules26082328

**Published:** 2021-04-16

**Authors:** Xia Gao, Xiangzhou Yi, Zhongyuan Liu, Xiuping Dong, Guanghua Xia, Xueying Zhang, Xuanri Shen

**Affiliations:** 1Hainan Engineering Research Center of Aquatic Resources Efficient Utilization in South China Sea, Hainan University, Hainan 570228, China; gaoxia20202021@163.com (X.G.); 201671435@yangtzeu.edu.cn (X.Y.); liuzhongyuan999@126.com (Z.L.); xiaguanghua2011@126.com (G.X.); 2College of Food Science and Technology, Hainan University, Hainan 570228, China; 3Collaborative Innovation Center of seafood Deep Processing, Dalian Polytechnic University, Dalian 116000, China; dxiuping@163.com; 4Key Laboratory of Seafood Processing of Haikou, Hainan 570228, China

**Keywords:** curcumin, fish head phospholipid, liposomes, inflammation

## Abstract

In this study, we compared the characteristics and in vitro anti-inflammatory effects of two curcumin liposomes, prepared with golden pompano head phospholipids (GPL) and soybean lecithin (SPC). GPL liposomes (GPL-lipo) and SPC liposomes (SPC-lipo) loaded with curcumin (CUR) were prepared by thin film extrusion, and the differences in particle size, ζ-potential, morphology, and storage stability were investigated. The results show that GPL-lipo and SPC-lipo were monolayer liposomes with a relatively small particle size and excellent encapsulation rates. However, GPL-lipo displayed a larger negative ζ-potential and better storage stability compared to SPC-lipo. Subsequently, the effects of phospholipids in regulating the inflammatory response of macrophages were evaluated in vitro, based on the synergistic effect with CUR. The results showed that both GPL and SPC exerted excellent synergistic effect with CUR in inhibiting the lipopolysaccharide (LPS)-induced secretion of nitric oxide (NO), reactive oxygen species (ROS), and pro-inflammatory genes (tumor necrosis factor (TNF)-α, interleukin 1β (IL-β), and interleukin 6 (IL-6)) in RAW264.7 cells. Interestingly, GPL-lipo displayed superior inhibitory effects, compared to SPC-lipo. The findings provide a new innovative bioactive carrier for development of stable CUR liposomes with good functional properties.

## 1. Introduction

Inflammation is a defensive immune response that is generated by an innate immune system response to endogenous or exogenous stimuli, such as dead cells, pathogens, and irritants. It is closely associated with many diseases, such as diabetes, arthritis, obesity, and cancer [1]. Suppressing inflammation can greatly inhibit the development of these diseases. The clinical use of traditional nonsteroidal anti-inflammatory drugs is associated with adverse gastrointestinal or cardiovascular effects, limiting their use and development. Therefore, screening for anti-inflammatory substances from natural products with low side effects and abundant sources, has become a major direction in food and pharmaceutical development.

Curcumin (1,7-bis(4-hydroxy-3-methoxyphenyl)-1,6-heptadiene-3,5dione) (CUR) is a widely studied hydrophobic polyphenol that is claimed to have numerous health effects, such as anti-oxidant, anti-inflammation, and anti-tumor activities, especially the inhibitory effect on inflammatory responses. Zhu et al. [2] reported that CUR promoted the degradation of inflammatory factors expressed by lipopolysaccharide (LPS)-stimulation. Unfortunately, the low water solubility, poor chemical stability, and low oral bioavailability of CUR have become a challenge for its development and application. To address these challenges, the solubility of CUR has increased via a variety of drug delivery systems, including nanoparticles, liposomes, micelles, self-assembled microemulsions, and solid dispersions [3,4,5]. Using this strategy, significant results have been achieved in basic structure research, nanobiotechnology, and nanomedicine [6]. However, most studies focused on active substance solubility and bioavailability enhancement, with little attention paid to the interaction between the carrier and the active substances, in terms of physiological activity.

Liposomes with colloidal and vesicular structures are composed of one or more lipid bilayers. Liposomes are carriers for drugs with poor bioavailability and water insolubility, and they are widely used in drug delivery systems, such as proteins, peptides, antibiotics, anticancer agents, and polyphenols [7]. The vesicles are biocompatible and biodegradable, and their structure is similar to cell membranes [6]. Thus, liposomes have become suitable and popular drug carrier alternatives for improving drug pharmacodynamics or therapeutic effects. Soybean lecithin and egg yolk lecithin are traditional liposomal materials. These phospholipids contain high degrees of unsaturation and, thus, have higher susceptibility to oxidation, as well as lower physical and storage stability [8]. Marine phospholipids contain phosphatidylcholine, phosphatidylethanolamine, and phosphatidylinositol like those in soybean lecithin. In addition, they contain large amounts of sphingomyelin and long-chain polyunsaturated fatty acids (docosahexaenoic acid, eicosapentaenoic acid) [9]. Therefore, marine phospholipids are some of the best potential candidates for the manufacture of liposomes. Phospholipids themselves have multiple physiological activities, so they may provide additional benefits to liposomes. It reported that phosphatidylcholine supplementation enhanced fatty acid β-oxidation and reduced liver fat content, which is a beneficial effect that could enhance the hepatoprotective effect of CUR in patients with nonalcoholic fatty liver disease or steatohepatitis [10,11]. Therefore, it is reasonable to speculate that there may be a potential synergistic effect between CUR and phospholipids in terms of inflammation. Golden pompano, one of the most valuable marine fish in Southeast Asian countries, has good nutritional values and is very popular among consumers. Fish head is one of the main processing by-products, and the phospholipids in it account for about 14.6% of the total fat [12]. Golden pompano head phospholipid (GPL) contains a number of polyunsaturated fatty acids, which have promising physiological activities, such as anti-inflammatory and anti-tumor activities [13,14]. GPL may have a greater potential for physiological activity, compared to soy lecithin.

In this study, GPL and SPC were employed to prepare GPL liposomes (GPL-lipo) and SPC liposomes (SPC-lipo), respectively, by a thin film extrusion method. The liposomes were used to encapsulate CUR and were then characterized and compared in terms of particle sizes, ζ-potential, morphology, and storage stability. After that, the effects of co-treatment of CUR with GPL/SPC on LPS-induced NO secretion in RAW264.7 cells was investigated. Finally, based on the experimental results, the GPL-lipo and SPC-lipo were functionally evaluated in terms of in vitro anti-inflammatory properties.

## 2. Results

### 2.1. Characterization of Liposomes

#### 2.1.1. Particle Size/ζ-Potential/Encapsulation Efficiency

As summarized in Table 1, before extrusion, the particle sizes of empty GPL, empty SPC, GPL-lipo, and SPC-lipo were 834.07 ± 25.72, 788.87 ± 11.52, 596.4 ± 26.63, and 787.83 ± 7.38 nm, respectively. After extrusion, the particle sizes decreased to 155.66 ± 1.10, 170.60 ± 1.69 nm. All four liposomes were negatively charged liposomes. Among them, GPL-lipo exerted the largest absolute potential, indicating that GPL-lipo was more stable than the other three samples. Liposomes prepared in this study had a satisfactory encapsulation capacity for CUR, with encapsulation efficiency reaching up to 85%. This result is similar to that of a previous study (about 82%) [15].

#### 2.1.2. Transmission Electron Microscopy (TEM)

TEM was used to study the morphology and accurately estimate the size of these liposomes formed. Representative TEM images of GPL-lipo and SPC-lipo show that the liposomes were spherical particles (Figure 1). Particle diameters were between 100 and 200 nm, which were in good agreement with the particle size data (Table 1).

#### 2.1.3. Storage Stability

GPL-lipo and SPC-lipo were kept in the refrigerator to monitor the change rate of the particle size and polydispersity index (PDI) over four weeks of storage. As depicted in Figure 2A, the particle size of GPL-lipo and SPC-lipo increased from 155.66 ± 1.10 nm and 157.97 ± 1.41 nm to 175.90 ± 1.84 nm and 205.37 ± 1.07 nm, respectively, after 28 d of storage. The particle size of GPL-lipo increased gradually during storage, while the particle size of SPC-lipo increased sharply from day 7 and then plateaued after day 14. It was found that the PDI values of GPL-lipo and SPC-lipo increased from 0.12 ± 0.03 and 0.17 ± 0.02 to 0.33 ± 0.05 and 0.39 ± 0.02, respectively, with increasing storage time (Figure 2B). This indicates that the system gradually became unstable. Overall, GPL-lipo showed better stability than that of SPC-lipo [16].

### 2.2. In Vitro Evaluation of the Anti-Inflammatory Activity of Two Liposomes

#### 2.2.1. Cell Viability

First, the effects of samples on the proliferation rate of RAW264.7 cells were determined by MTT assay. As shown in Figure 3, compared with the control group, treatment with GPL for 24 h had no toxic effects on RAW264.7 cells (Figure 3A). CUR (Figure 3B), CUR and GPL/SPC co-treatment and GPL-lipo/SPC-lipo (Figure 3C) significantly promoted the growth of RAW264.7 cells. Among them, GPL-lipo exhibited the best growth-promoting effect. The results suggest that they could be further used for subsequent experiments.

#### 2.2.2. Nitric Oxide (NO) Production

Increased production of NO is typical in LPS-stimulated macrophages; therefore, it is necessary to evaluate NO production. Since NO is readily oxidized in the biological environment, nitrite, a stable product of NO, can be used to indirectly measure NO production. As shown in Figure 4, it was concluded that CUR, GPL, and SPC groups substantially inhibited the release of NO, but CUR and GPL/SPC co-treatment groups exhibited better effects. Inspired by these results, we investigated the liposome formulations. The liposomes exhibited the most obvious inhibitory effects, likely because the formation of liposomes promotes the absorption of CUR. It is worth mentioning that the inhibitory effect of GPL-lipo (25.83%) was higher than that of SPC-lipo (17.21%), which was comparable with the dexamethasone (DX) treatment group (28.10%, positive control).

#### 2.2.3. Reactive Oxygen Species (ROS) Expression

GPL and SPC served as encapsulation carriers to effectively increase the inhibitory effect of CUR on NO production in RAW264.7 cells. Therefore, we next focused on whether the two carriers had effects on CUR in suppressing the production of ROS and inflammatory factors. As shown in Figure 5, the ROS fluorescence intensity of LPS-induced RAW264.7 cells was significantly higher than that of the control group (*p* < 0.01). After treatments with CUR, GPL-lipo, and SPC-lipo, a marked decrease in the LPS-stimulated ROS level was observed (*p* < 0.01). However, it should be noted that the inhibitory effect of CUR on the ROS production was enhanced when encapsulated in liposomes. Additionally, the GPL-lipo treatment group showed the best inhibitory effect (40.85%), which was 1.52 times stronger than that of the SPC-lipo group (26.81%).

#### 2.2.4. Quantitative Reverse Transcription-PCR (RT-qPCR)

The regulatory effects of GPL-lipo and SPC-lipo on inflammatory factors, such as tumor necrosis factor (TNF)-α, interleukin 1β (IL-1β), and interleukin 6 (IL-6) at the gene level, were explored by qRT-PCR. As indicated in Figure 6, the expression of TNF-α, IL-1β, and IL-6 significantly increased after LPS treatment (1.0 mg/mL), while CUR, GPL-lipo, and SPC-lipo effectively downregulated the over-expression of these inflammatory cytokines. Interestingly, GPL-lipo greatly inhibited the expression of TNF-α (78.79%, Figure 6A), and the inhibition effect was even more remarkable than that of the DX group (64.40%).

## 3. Discussion

CUR had shown activity against many inflammatory diseases, such as arthritis [17] and inflammatory bowel disease [18]. However, the poor water solubility of CUR limits its physiological effects and current research has suggested many possible ways to rationalize its efficacy [19,20]. Liposomes are strategic technological tools that have been used in recent years to improve the bioavailability of CUR [21]. This is due to the similarity of the phospholipid bilayer of liposomes to cell membranes, which makes them suitable carriers for transporting biomolecules through human tissues.

In recent years, numerous studies have shown that the physiological activity of marine phospholipids is superior to that of SPC. Following these evidences, in the present study, CUR liposomes were prepared using GPL/SPC as wall materials. To evaluate the effect of the encapsulation of CUR on the vesicle arrangement, empty liposomes were also prepared and characterized. Before extrusion, the particle size of the empty GPL liposome (834.07 ± 25.72 nm) was equivalent to that of the empty SPC (788.87 ± 11.52 nm), but a slight difference was observed after CUR was loaded. The particle size of SPC-lipo was 787.83 ± 7.38 nm, which was basically unchanged compared to the empty GPL. However, the particle size of GPL-lipo decreased to 596.40 ± 26.63 nm. It was reported that the phosphatidylcholine contents in SPC and GPL were different, and the size of the hydrophilic head group in the GPL is relatively large [22]. This might explain the difference in the particle size between GPL-lipo and SPC-lipo. To enhance the permeability and retention (EPR) effect of the liposomes, an extruder was used to reduce their particle size [23]. After extrusion, the particle sizes of GPL-lipo and SPC-lipo were less than 200 nm. The absolute values of the ζ-potential of GPL-lipo and SPC-lipo were larger, compared to the empty liposomes. This might be due to the fact that encapsulation of CUR caused changes in the structure of the liposome surface, resulting in a change in the orientation of the phosphatidylcholine head group on the liposome surface and leading to the measured negative ζ-potential [24]. In addition, the absolute ζ-potential of GPL-lipo was significantly higher than that of SPC-lipo (35.47 ± 0.87 vs. 25.30 ± 0.56 mV), which might be caused by the difference in phospholipid composition between GPL and SPC. Both GPL and SPC had negatively charged lipid phosphatidylethanolamine, which contributes to the anionic nature of liposomes [25]. Therefore, the higher phosphatidylethanolamine content in GPL led to a higher ζ-potential. Traditionally, the absolute value of the ζ-potential greater than 30 mV is considered a high potential, which will generate a large electrostatic repulsion between liposome particles, thereby improving the stability of the liposome system [26]. Therefore, GPL-lipo might be more stable than SPC-lipo, and this was confirmed in the storage stability experiments.

The inflammatory response is the primary mechanism by which the host fights infection. During inflammation, macrophages residing in human tissues and organs are activated by infectious agents or injury. Activated macrophages produce inflammatory mediators, such as NO, IL-1β, and TNF-α, to defend against invading pathogens. However, when inflammatory mediators are overproduced, they lead to dysregulation of inflammation, which is a central pathological process in various disease states. Therefore, inhibition of the excessive production of pro-inflammatory mediators should be a viable approach to treat these diseases [27].

NO is an important signaling molecule in vivo, catalyzed by inducible nitric oxide synthase (iNOS) for levorotatory arginine production. iNOS is a carrier of intercellular gas messaging and is activated in the inflammatory response, increasing iNOS expression and catalyzing NO production. Excess NO can induce the development and progression of inflammatory diseases. Therefore, effective inhibition of overproduction of NO is one of the important measures to control the inflammatory response [28,29]. In this study, CUR significantly reduced NO secretion, and this is consistent with a previous report [30]. Then, we found that CUR and GPL/SPC have an additive effect on inhibiting NO secretion, which might be due to the anti-inflammatory activity of phospholipids and their emulsifying effect. Together, these properties promote the absorption of CUR [31,32,33]. Finally, the experimental results of NO inhibition by both liposomes show that GPL-lipo and SPC-lipo not only inhibited NO secretion in RAW264.7 cell, but their inhibition was also superior to that of CUR and GPL/SPC co-treatment groups. This was attributed to the fact that the prepared liposomes were negatively charged, and their structures were very similar to the cell membrane structure, allowing liposomes to easily enter the cells and deliver CUR to the cells.

ROS is essential for host defense and is produced in the mitochondria of phagocytes when stimulated by microbes and inflammation [34]. ROS is a key signaling molecules that plays an important role in signal transduction. Excessive ROS can lead to the upregulation of iNOS expression and the induction of inflammatory cytokines by erythropoietin. In addition, ROS can act as a secondary messenger to further amplify the inflammatory response induced by LPS [35]. Therefore, the regulation of ROS is an important target to identify the mechanism of inflammation induction. Compared with the LPS group, GPL-lipo and SPC-lipo groups significantly reduced the level of ROS, and their effects were superior to the CUR group. The results of qRT-PCR showed that GPL-lipo and SPC-lipo also reduced the expression of inflammatory genes (TNF-α, IL-1β, and IL-6 gene) induced by LPS in RAW264.7 cells. TNF-α is a pro-inflammatory cytokine produced by a large number of immune cells during inflammation. It can upregulate many genes expressed in adipose tissue, such as IL-6, IL-1β, and angiotensinogen. These genes are involved in inflammatory and immune responses [36]. Our study focused on whether the effect of GPL-lipo in regulating the inflammatory response of macrophages was superior to that of equal amounts of CUR and SPC-lipo. The results confirmed that GPL-lipo showed significant advantages over CUR and SPC-lipo. This was attributed to the abundance of polyunsaturated fatty acids in GPL and the role of the acyl chain composition of GPL on the permeability of the monolayer [37]. Similar reports were reported by Du et al. [9], and the authors found that liposomes consisting of phospholipids from sea stars and sea cucumbers showed significantly higher translocation and uptake efficiency than SPC liposomes, suggesting that marine complex lipid liposomes have higher uptake efficiency in small intestinal epithelial cells.

## 4. Materials and Methods

### 4.1. Materials

SPC was purchased from Shanghai Enzyme-linked Biotechnology Limited Company (Shanghai, China). CUR, cholesterol (Chol), and DX were purchased from Aladdin Chemicals (Shanghai, China). Chloroform and anhydrous ethanol were purchased from Guangdong Xilong Technology Co (Guangdong, China). Mouse RAW264.7 cells were purchased from the Cell Resource Center (Shanghai Institute of Biological Sciences, Chinese Academy of Sciences, Shanghai, China). LPS and 3-(4,5-dimethyl-2thiazolyl)-2,5-diphenyl-2-H-tetrazolium bromide (MTT) were obtained from Sigma Aldrich Trading Co., Ltd. (Shanghai, China). Dulbecco was modified with Eagle’s medium (DMEM), phosphate-buffered saline (PBS), dimethyl sulfoxide (DMSO), and fetal bovine serum (FBS) were purchased from Gibco Life Technologies (Carlsbad, CA, USA). The NO detection kit and the ROS kit were obtained from Shanghai Biyuntian Biotechnology Co., Ltd. (Shanghai, China).

### 4.2. Extraction of GPL

GPL was extracted according to the method of Chen et al. [38], with slight modification. Fish heads were homogenized, extracted with 95% ethanol (1:4, *v*/*v*), and sonicated at 25 °C for 1.5 h (×3) (power 60 W). Subsequently, impurities such as pigments were removed with chloroform/methanol/H_2_O/ethanol (8:4:3:3, *v*/*v*), and the organic phase was collected. To the organic phase, 0.2 × the volume of 0.9% (m/v) NaCl solution was added to remove protein. The organic phase was collected and spun dry to obtain total lipid. Finally, the total lipid was passed through a silica gel column (silica irregular 40–60 μm 60 A 12 g) (Tianjin Bona Ijar Technology Co., Tianjin, China) to separate GPL. The GPL was stored at −20 °C before use.

### 4.3. Preparation of Liposomes

The CUR liposomes were prepared with GPL or SPC. Briefly, 75 mg of phospholipids (GPL/SPC), 15 mg of Chol, and CUR (0 or 5 mg) were dissolved in 5 mL chloroform, followed by spin-drying at 55 °C using a spin evaporator (Shanghai Yarong Biochemical Instrument Factory, Shanghai, China) to form a film. Then, PBS (0.05 mol/L, pH 7.4) was added for hydration to form a liposome solution. The hydrated solution was filtered sequentially by passing through 400 nm, 200 nm, and 100 nm polycarbonate membranes, using an extruder (Avanti Mini Extruder; Alabaster, AL, USA) to obtain uniformly sized liposomes. GPL-lipo and SPC-lipo were stored in small amber bottles at −4 °C.

### 4.4. Measurement of Particle Size Distribution, PDI, ζ-Potential, and Encapsulation Efficiency

The mean particle size distribution, PDI, and ζ-potential of liposomal vesicles were determined using a dynamic light scattering (DLS) instrument Nano ZS (Malvern Instruments Ltd., Malvern, UK). The encapsulation efficiency of CUR was determined by a double-beam UV-vis spectrophotometer (Beijing Puxi General Instrument Co., Ltd., Beijing, China). After the preparation of liposomes, the samples were centrifuged (Sartorius Group., Göttingen, Germany) at 3500 rpm for 15 min at 25 °C. The supernatant was collected after centrifugation. The vesicles mixed with ethanol in the same volume, then broken by sonication at 25 °C for 10 min and fixed with 10 mL anhydrous ethanol.

The absorbance was measured at 425 nm. The mass of encapsulated CUR was calculated from the standard curves prepared from different concentrations of CUR solutions. In the same way, another sample of the same volume was taken from the same sample and directly sonicated to break the liposome membrane to detect the mass of total CUR. The calculation of encapsulation efficiency was based on Equation (1):(1)$$Encapsulation\;Efficiency=\frac{m_{1}}{m_{2}}\times 100\%$$
where *m_1_* is the amount of CUR in the liposome, and *m_2_* is the amount of total CUR.

### 4.5. TEM

The liposomal vesicle morphology was observed by JEM-1200EX (TEM, Japanese Electronics Co., Ltd., Tokyo, Japan). Liposomes were diluted prior to imaging and then transferred to a 200-mesh carbon-coated copper grid. The samples were negatively stained by phosphotungstic acid solution (3%) for 90 s and air-dried at 25 °C for 2 h. Once dry, the copper grid was loaded into the TEM and viewed at 40 kV, with a 5000–20,000 magnification.

### 4.6. Storage Stability

To eliminate microbial effects, 0.02% sodium azide was added to 20 mL of liposomes in brown bottles and stored at 4 °C for 28 d. At the indicated times (0, 7, 14, 21, and 28 days), 3 mL of each sample were collected to determine the particle size and PDI. Each sample was measured in triplicate.

### 4.7. Cell Viability Assay

RAW 264.7 macrophages were cultured in DMEM (medium supplemented with 10% FBS) in a humidified atmosphere of 5% CO_2_–95% air at 37 °C. The cytotoxicity of the samples to macrophages was determined by MTT assay. The samples were divided into three groups: CUR group (0, 0.125, 0.25, 0.5, and 1 μg/mL); GPL group (0, 6.25, 12.5, 25, and 50 μg/mL); and CUR (1 μg/mL) and GPL/SPC (50 μg/mL) co-interaction group. GPL-lipo/SPC-lipo (50 μg/mL, using phospholipid concentration) was used as the control group. The RAW264.7 macrophage cells were seeded in a 96-well plate at a density of 1.5 × 10^5^ cells/well and grown for 24 h. Then, cells were treated for 24 h and reacted with 20 µL MTT (5 mg/mL) for 4 h. The supernatant was discarded, and 150 μL of DMSO were added to each well. The absorbance was measured at 490 nm by an enzyme-linked immunosorbent-assay reader (BioTek Epoch, Thermo Fisher Scientific, Waltham, MA, USA). The cell cytotoxicity was calculated using Equation (2):(2)$$Cell\;viability\%=\frac{ODsample-ODblank}{ODcontrol-ODblank}\times 100\%$$
where *OD* denotes the optical density; *ODcontrol* represents *OD* with 100% survival; and *ODblank* represents *OD* with no cells.

### 4.8. NO Production Assays

NO concentration was determined based on the accumulation of nitrite, a major stable product of NO, using a Griess reagent-based colorimetric assay [39]. RAW 264.7 cells (1.5 × 10^5^ cells/well) were plated in 96-well plates and treated with DX (20 μg/mL); CUR (1 μg/mL); GPL/SPC (50 μg/mL); CUR (1 μg/mL) and GPL/SPC (50 μg/mL) co-treatment, and GPL-lipo/SPC-lipo (50 μg/mL, using phospholipid concentration) as the control. NO production was stimulated by incubating with LPS (1 μg/mL) for 24 h. Then, the culture medium was mixed with the Griess reagent at ratio of 1:1 (*v*/*v*), and incubated at 25 °C for 15 min. The absorbance was measured at 540 nm using an enzyme-linked immunosorbent-assay reader. The nitrite levels of the media were calculated from regression analysis using known concentrations of sodium nitrite to generate a standard curve.

### 4.9. ROS Production Assays

The effect of DX (20 μg/mL), CUR (1 μg/mL), GPL-lipo, and SPC-lipo on ROS secretion by RAW264.7 cells were determined using a ROS assay kit. DCFH-DA (2′,7′-dichlorodihydrofluorescein diacetate) was diluted with serum-free medium at 1:1000, and the final concentration was 10 μmol/L. Cells were cultured in 48-well plates. After 24 h, the cell culture medium was removed, and 500 μL diluted DCFH-DA were added. Cells were incubated and washed according to the kit instructions. The cells were resuspended with PBS and collected into 1.5 mL tubes. The ROS content was determined by flow cytometry (FCM, Beckman-Kurt Co., Ltd., CA, USA).

### 4.10. RT-qPCR

According to the manufacturer’s instructions, RNAiso Plus (TAKARA, China) was used to isolate total RNA from RAW264.7 cells. cDNA from isolated RNA was prepared with the ReverTra Ace qPCR RT Master Mix (Toyobo, Tokyo, Japan). Quantitative reverse transcription-PCR was performed using the SYBR Green Realtime PCR Master Mix (Toyobo, Tokyo, Japan), according to the manufacturer’s protocol. PCR primer sequences are listed in Table 2. Relative gene expressions were normalized to GAPDH via 2^−ΔΔCT^ method [40,41].

### 4.11. Statistical Analysis

The data were processed by Origin85 (OriginLab Corporation, Northampton, MA, USA), SPSS Statistics17.0 (SPSS Inc., Chicago, IL, USA), and the data were expressed as mean ± SD, using one-way ANOVA for statistical comparison between groups. After Duncan multi-range test, *p* < 0.05 means there were significant differences among the samples, *p* < 0.01 means there were extremely significant differences among the samples.

## 5. Conclusions

Although GPL-lipo and SPC-lipo had a similar encapsulation efficiency, GPL-lipo exhibited a higher absolute ζ-potential, stronger binding capacity to CUR, and better stability than SPC-lipo. Synergistic inhibition on NO caused by CUR and GPL was higher than that of CUR and SPC. In addition, compared to CUR and SPC-lipo, GPL-lipo exhibited the strongest inhibitory effect on LPS-induced NO, and GPL-lipo decreased the expression of inflammatory genes (TNF-α, IL-1β, and IL-6) and ROS secretion in RAW264.7 cells. The results imply that liposomes can improve the anti-inflammatory activity of CUR and are excellent drug carriers. Meanwhile, GPL-lipo had the best anti-inflammatory activity, which indicate that marine phospholipids are more suitable as liposome wall materials. As a next step, animal studies and well-designed human intervention studies should be performed to demonstrate the improved bioavailability, as well as the efficacy and safety of GPL-lipo. Based on this work, marine phospholipid liposomes could be developed as a functional food to mitigate chronic inflammation.

## Figures and Tables

**Figure 1 molecules-26-02328-f001:**
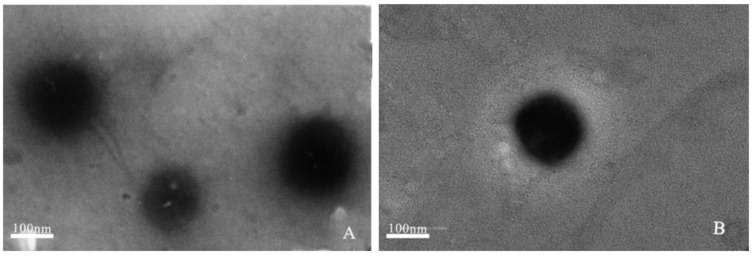
TEM images of GPL-lipo (**A**) and SPC-lipo (**B**).

**Figure 2 molecules-26-02328-f002:**
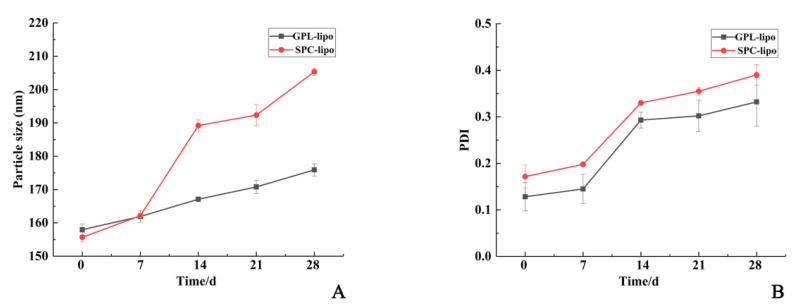
Comparison of storage stability of GPL-lipo and SPC-lipo. Comparative results of the change in encapsulation efficiency during storage for GPL-lipo and SPC-lipo are shown in the subfigures (**A**) and (**B**). The values reported are averaged over three measurements.

**Figure 3 molecules-26-02328-f003:**
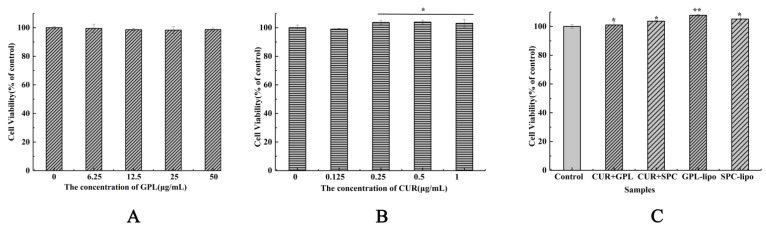
Effects of GPL (0–50 μg/mL) (**A**), CUR (0–1 μg/mL) (**B**), CUR (1 μg/mL) and GPL/SPC (50 μg/mL) co-treatment and liposome (50 μg/mL, using phospholipid concentration as standard) (**C**) on cell viability of RAW264.7 cells. Data are expressed as mean ± SD (*n* = 4) and multiple comparisons were done using one-way ANOVA analysis. * *p* < 0.05; ** *p* < 0.01 versus control group. Abbreviations: GPL-golden pompano head phospholipids, SPC-soybean lecithin, CUR- curcumin.

**Figure 4 molecules-26-02328-f004:**
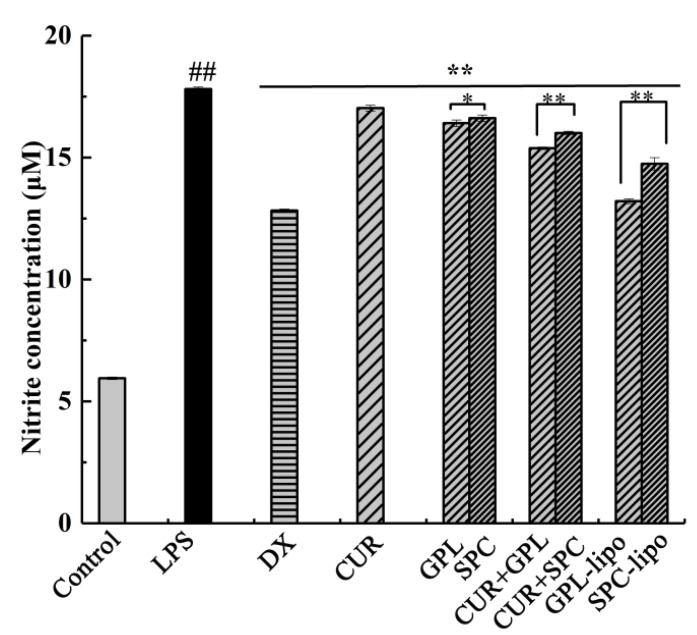
Effects of DX (20 μg/mL), GPL (50 μg/mL), CUR (1 μg/mL), CUR (1 μg/mL) and GPL/SPC (50 μg/mL) co-treatment and liposome (50 μg/mL, using phospholipid concentration as standard) on the NO level in RAW264.7 cells. Data are expressed as mean ± SD (*n* = 4) and multiple comparisons were done using one-way ANOVA analysis. ^##^
*p* < 0.01 versus control group; * *p* < 0.05; ** *p* < 0.01 versus the LPS group. Abbreviations: DX-dexamethasone, GPL-golden pompano head phospholipids, SPC-soybean lecithin, CUR-curcumin, NO- nitric oxide, LPS-lipopolysaccharide.

**Figure 5 molecules-26-02328-f005:**
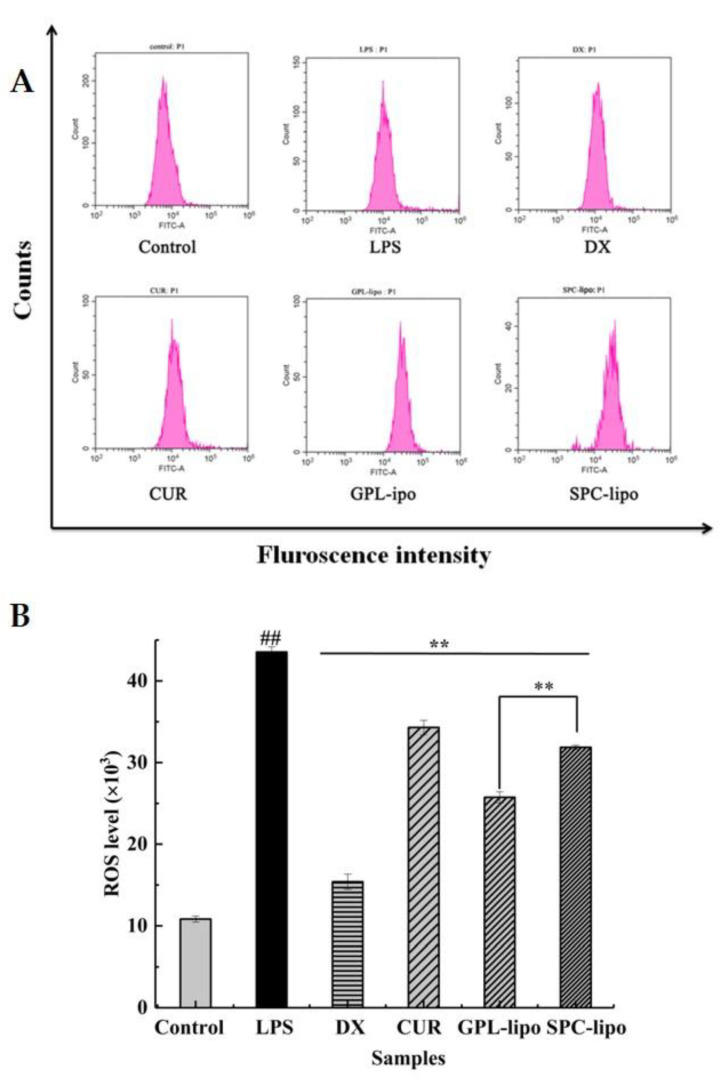
Effects of GPL-lipo and SPC-lipo on the reactive oxygen species (ROS) level in RAW264.7 cells (**A**,**B**). Data are expressed as mean ± SD (*n* = 4) and multiple comparisons were done using one-way ANOVA analysis. ^##^
*p* < 0.01 versus control group; ** *p* < 0.01 versus the LPS group. Abbreviations: DX-dexamethasone, GPL-golden pompano head phospholipids, SPC-soybean lecithin, CUR-curcumin, NO- nitric oxide, LPS-lipopolysaccharide.

**Figure 6 molecules-26-02328-f006:**
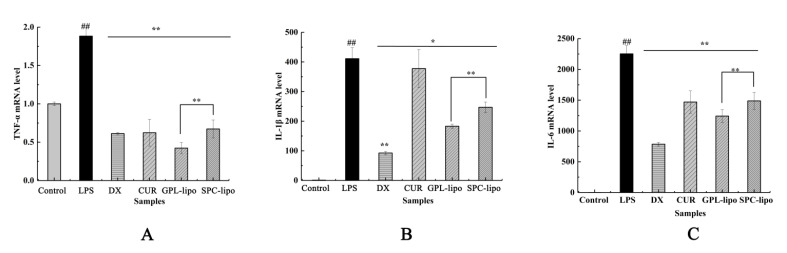
Effects of GPL-lipo and SPC-lipo on the expression of tumor necrosis factor (TNF)-α (**A**), interleukin 1β (IL-1β) (**B**), and interleukin 6 (IL-6) (**C**) in RAW246.7 cells. Data are expressed as mean ± SD (*n* = 4) and multiple comparisons were done using one-way ANOVA analysis.; ^##^, *p* < 0.01 versus the control group; *, *p* < 0.05; **, *p* < 0.01 versus the LPS group.

**Table 1 molecules-26-02328-t001:** Comparison of particle size, ζ-potential, and encapsulation efficiency of two CUR liposomes.

Liposomes Formulations	Particle Size (nm)		ζ-Potential (mV)	Encapsulation Efficiency (%)
	Pre-Extrusion	Pro-Extrusion		
Empty GPL liposome	834.07 ± 25.72	170.60 ± 1.69	−28.23 ± 1.46	-
Empty SPC liposome	788.87 ± 11.52	169.83 ± 0.6	−22.33 ± 0.86	-
GPL-lipo	596.4 ± 26.63	155.66 ± 1.10	−35.47 ± 0.87	86.72 ± 2.74
SPC-lipo	787.83 ± 7.38	157.97 ± 1.41	−25.30 ± 0.56	89.25 ± 3.20

Values are the mean of triplicate analyses ± SD.

**Table 2 molecules-26-02328-t002:** RT-qPCR primer sequence.

Genes	Forward	Reverse
GAPDH	CATGTTCCAGTATGACTCCACTC	GGCCTCACCCCATTTGATGT
TNF-α	TATGGCTCAGGGTCCAACTC	GGAAAGCCCATTTGAGTCCT
IL-1β	GTTGACGGACCCCAAAAGAT	CCTCATCCTGGAAGGTCCAC
IL-6	CACGGCCTTCCCTACTTCAC	TGCAAGTGCATCATCGTTGT

## Data Availability

The data presented in this study are available upon request from the corresponding author.

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
