# Peer review of "Comparative Study on Curcumin Loaded in Golden Pompano (Trachinotus blochii) Head Phospholipid and Soybean Lecithin Liposomes: Preparation, Characteristics and Anti-Inflammatory Properties"

_molecules, 2021, doi:10.3390/molecules26082328_

Round 1

Reviewer 1 Report

Manuscript number: 1150755

Title: Comparative study on curcumin loaded in Golden Pompano (Trachinotus blochii) head phospholipid and soybean lecithin liposomes: Preparation, characteristics and anti-inflammatory properties.

I like the effort of the authors to evaluate the characteristics of two types of liposomes to deliver curcumin and their anti-inflammatory effect. The large number of biological properties of curcumin are widely known, but its low stability, solubility and availability reduce the application of this compound. Thus, in my opinion, developing new (and sustainable) strategies to enhance these drawbacks is quite important.

In the following paragraphs, I will provide clear information in order to improve the manuscript. In general, the English level is good and I cannot see many discrepancies. In general, I would recommend to check the abbreviations used along the manuscript.

ABSTRACT

The abbreviation of CUR is missing.

INTRODUCTION

The introduction is clear, fluid and gives the necessary context for the article (inflammation, curcumin properties and limitations, liposome strategies). The objectives have been also exposed clearly. In my opinion, this section has been correctly performed.

RESULTS

In my opinion, authors have presented their results in a clear and precise way. This section has been correctly performed. In general, Table 1 presents relevant information and Figures are visual and easy to understand. I have detected a few mistakes:

  • Line 99: the abbreviation of curcumin was stated before.
  • Line 114: the abbreviation of PDI is missing.
  • Line 131: CURand, please correct.
  • In Figure 5, the arrow above the fluorescence intensity is not straight, please change.

DISCUSSION

Authors have discussed their results properly, comparing the findings with previous studies, when possible. In this section, I have some comments for the authors.

  • Have GPL and SPC liposomes employed with other bioactive compounds?
  • I miss a comparison between the liposomes used in this study with other systems used for curcumin’ delivery. Are they less/more effective?
  • Line 217: the abbreviation PE has not been stated before.
  • Line 258: A reference or references about the previous studies are missing.

MATERIAL AND METHODS

From my point of view, the scientific design and methodology employed is suitable for achieve the objectives proposed.

  • Line 314: the abbreviation EE has not been stated before.

CONCLUSIONS

Conclusions coincide with the results of the study and summarized them well, highlighting the suitability of GPL-liposomes to encapsulate curcumin. In addition, they have considered subsequent studies in the evaluation of these liposomes. However, I would suggest the authors to make more extensive and complete conclusions.

  • Line 381: thestrongest, please correct.

FINAL CONCLUSIONS

In my opinion, authors have performed an extensive work, with correct scientific bases and which provides interesting results. However, discussion could be deeper, so I would suggest MAJOR REVISIONS of the manuscript to improve its quality.

Author Response

Dear  Reviewer:

Thank you for  for the your comments concerning our manuscript entitled “Comparative study on curcumin loaded in Golden Pompano (Trachinotus blochii) head phospholipid and soybean lecithin liposomes: Preparation, characteristics and anti-inflammatory properties” (ID: molecules-1150755). According to your comments, we have substantially revised our manuscript. Revised portions are marked with yellow shading in the paper. You can kindly find the point-to-point responses  in the attachment.

With thanks! 

Yours Sincerely,

Xia Gao

Reviewer 2 Report

In the manuscript of Gao et al. two curcumin-loaded liposomes made of different kind of phospholipids are compared regarding their size, stability and anti-inflammatory properties. The content of this manuscript is related to the scope of Molecules but it could not be accepted for publication in the present form.

The manuscript is really poorly written and it needs to be rewritten with the help of an English native or expert in this language.

Pag.2 line 49 “the water insolubility of CUR” Does the term insolubility need to be substituted with solubility

Pag.2 line 57 “liposomes are poorly bioavailable” perhaps the authors intend highly bioavailable?

Pag.2 line 70-71 reference 10 sounds wrong because it's doesn't report the effect of phosphatidylcholine supplementation

Pag.2-3 lines 91-101The discussion about the effect of extrusion is trivial and in my opinion can be eliminated; in fact, the need to resize liposomes by means of extrusion or sonication is well known the and it is widely described in literature.

Pag.3 lines 106-107 “Representative TEM images of GPL-lipo and SPC-lipo showed that the 106 liposomes were spherical particles with monolayer structure (Figure 1).” The authors can not claim that the liposomes are unilamellar by the TEM technique employed. In order to determines the number of layers in the liposomes freeze-fracture TEM is necessary.

Pag.7 line 208 “the TEM results demonstrated that the liposomes obtained by extrusion were unilamellar liposomes”. The authors can not demonstrate the unilamellarity of liposomes. The sentence has to be eliminated.

Pag.9 it is not clear the description of the method employed in the Encapsulation Efficiency determination. Can the authors better explain? In my opinion they can not be sure that the liposomes are completely broken by sonication, a lysis by Triton could  better get the objective.

Pag.9. The storage stability of the formulation needs to be characterized not only by determining the liposomes size but also determining the curcumin release.

Author Response

(The authors gave the same response as above.)

Round 2

Reviewer 1 Report

Manuscript number: 1150755_v2

Title: Comparative study on curcumin loaded in Golden Pompano (Trachinotus blochii) head phospholipid and soybean lecithin liposomes: Preparation, characteristics and anti-inflammatory properties.

The authors have taken into consideration all the comments suggested previously and have carried out several changes to improve the overall quality of the manuscript.

ABSTRACT

The abstract has been modified according to the previous comment. In my opinion, the abstract is correct.

INTRODUCTION

In my opinion, this section is correctly performed, but I have some comments for the authors:

  • Line 69-70: I would suggest the authors to modified the phrases “Phospholipids themselves have multiple physiological activities. There may be additional benefits from phospholipids in liposomes”, to improve the fluidity.
  • Line 70: “However, it has been reported that phosphatidylcholine supplementation…” It is correct “However”? It seems that the authors are mentioning an example of the benefits of phospholipids in liposomes.
  • Line 72: “fat content, Which is a beneficial”, please correct.
  • Line 76: “is rich in nutrition”. I would recommend the authors to change the statement, to use a more correct terminology. For example, “it has good nutritional values”.

RESULTS, DISCUSSION

The authors have done all the modifications suggested in both sections. In addition, they have carried out several changes that increased the quality of the manuscript.

  • Line 147: “It is worth mentioning that the inhibitory effect of GPL-lipo (25.83%) was higher than that of SPC-lipo…”

CONCLUSIONS

The authors have considered the previous comment and they have extended and completed their conclusions. In my opinion, this section has been successfully modified.

  • 386: “GLP-lipo had the best…”

FINAL CONCLUSIONS

In my opinion, the authors have considered carefully all the suggestions. I have detected some minor mistakes, so I am suggesting MINOR REVISIONS before publication of the present work.

Author Response

Dear Reviewer:

We appreciate your effort in evaluating our manuscript entitled “Comparative study on curcumin loaded in Golden Pompano (Trachinotus blochii) head phospholipid and soybean lecithin liposomes: Preparation, characteristics and anti-inflammatory properties” (ID: molecules-1150755) and in providing constructive comments and suggestions. According to your comments, we have substantially revised our manuscript. Revised portions are marked with yellow shading in the paper. You can kindly find the point-to-point responses in the attachment.

With thanks!

Yours Sincerely,

Xia Gao
